# Research Progress and Management Strategies for the Common Mycotoxin Contamination of Traditional Chinese Medicines

**DOI:** 10.3390/jof11060411

**Published:** 2025-05-27

**Authors:** Zhimin Yang, Huali Xue, Ye Han, Hui Ding, Ying Zhang

**Affiliations:** 1Lanzhou Institutes for Food and Drug Control, Lanzhou 730050, China; yzmljh@163.com (Z.Y.); dhc467828@gmail.com (H.D.); 13609310016@163.com (Y.Z.); 2College of Science, Gansu Agricultural University, Lanzhou 730070, China; 3College of Food Science and Engineering, Gansu Agricultural University, Lanzhou 730070, China; hanyest1989@163.com

**Keywords:** traditional Chinese medicine, mycotoxin contamination, detection methods, regulatory recommendations

## Abstract

With rapid globalization and the increasing demand for traditional Chinese medicines, quality and safety has become a critical priority for both domestic and international markets. However, traditional Chinese medicines are susceptible to contamination by pathogenic microorganisms during the process of cultivation, growth, processing, storage, and transportation, which can lead to mycotoxin contamination that adversely affect the quality and safety of traditional Chinese medicines, and may pose potential threats to human health. This review summarizes mycotoxin contamination, the common detection methods, prevention and control measures, and regulatory recommendations, aiming to provide references for improving the quality standards and ensuring safety of these medications.

## 1. Introduction

Traditional Chinese medicines have been integral to health practices in China for over 2000 years. However, they are susceptible to contamination by pathogenic microorganisms during the process of cultivation, growth, processing, storage, and transportation, which lead to mycotoxin accumulation, posing potential health risks, including carcinogenicity, hepatotoxicity, and nephrotoxicity [1]. Mycotoxins are secondary metabolites produced by certain filamentous fungi under suitable temperature and humidity conditions [2,3]. As the world’s largest producer, consumer, and exporter of traditional Chinese medicines, China plays a pivotal role in the global medicine sales market. In 2022, China’s annual output of traditional Chinese medicinal materials exceeded 20 million tons, accounting for over 70% of global production. The domestic consumption scale reached approximately 1.1 trillion yuan in 2023. The export value of traditional Chinese medicine products (including Chinese medicinal materials, extracts, Chinese patent medicines and decoction pieces) amounted to $4.2 billion, with major market in Japan, South Korea, and Southeast Asia. Currently, the main exported medicinal materials, such as *Ginseng*, *Astragalus*, *Wolfberry*, *Angelica sinensis*, *Honeysuckle*, and *Licorice*, are highly sought after, yet the quality and safety of traditional Chinese medicines caused by mycotoxins has become a trade barrier. Although the Chinese Pharmacopoeia has updated and gradually included mycotoxin testing requirements for a variety of traditional Chinese medicines, as well as to add some detection methods and permissible limit values; however, the formulation of relevant standards and management measures are still lacking. Significant gaps remain in standardization and regulatory oversight.

In this review, we focus on the current situation of mycotoxin contamination in traditional Chinese medicines, summarize the types of mycotoxin contamination in different kinds of medicinal materials, and present mycotoxin detection methods and regulatory recommendations.

## 2. Mycotoxin Contamination in Traditional Chinese Medicines

With increasing attention being paid to the quality and safety of traditional Chinese medicines by consumers, the issue of mycotoxin contamination in traditional Chinese medicines has also garnered widespread concern. The contamination level of mycotoxins is related to the types of traditional Chinese medicines and the species of fungi. Mycotoxins are produced by certain species of *Aspergillus*, *Fusarium*, *Alternaria*, and *Penicillium* under appropriate temperature and humidity conditions during growth, storage, processing, and transportation. These fungi have the ability to produce more than 400 mycotoxins; however, only a few mycotoxins have been detected in traditional Chinese medicines, including aflatoxins (AFs), ochratoxins (OTs), zearalenone (ZEN), deoxynivalenol (DON, also known as vomitoxin), fumonisins (FBs), and patulin (PAT) [4,5,6,7,8]. These mycotoxins exhibit carcinogenic, teratogenic, mutagenic, and immunosuppressive effects [9,10], among others, and pose unpredictable threats to the health of animals and humans, even at low dose levels [11,12].

The Chinese Pharmacopoeia has been paying increasing attention to the types of traditional Chinese medicines that are susceptible to mycotoxin contamination. The 2010 edition of the Chinese Pharmacopoeia included aflatoxin testing requirements for the first time, covering five types of traditional Chinese medicines: *Bombyx batryticatus*, *Citri reticulatae pericarpium*, *Ziziphi spinosae semen*, *Persicae semen*, and *Sterculiae lychnophorae semen*. On this basis, the 2015 edition of the Chinese Pharmacopoeia added 14 types of AF testing requirements for traditional Chinese medicines (including *Jujubae fructus*, *Hirudo*, *Pheretima*, *Myristicae semen*, *Scorpio*, *Cassiae semen*, *Hordei fructus germinatus*, *Polygalae radix*, *Quisqualis fructus*, *Platycladi semen*, *Nelumbinis semen*, *Scolopendra*, *Arecae semen*, and *Coicis semen*). Further expanding on the 2015 edition, the 2020 edition added five types of traditional Chinese medicines, including *Corydalis rhizoma*, *Eupolyphaga steleophaga*, *Aspongopus*, *Vespae nidus*, and *Strychni semen*. The 2025 edition of the Chinese Pharmacopoeia further strengthens the control of mycotoxins in traditional Chinese medicines and decoction pieces. The aflatoxin testing requirements now include bran-fried *Coix seed* decoction pieces. Additionally, the zearalenone testing requirements, which previously applied to *Coicis semen* in the 2020 edition, have been expanded to include bran-fried *Coix seed* decoction pieces. Furthermore, ochratoxin A testing requirements have been introduced for the first time for medicinal materials such as *Astragalus radix* and *Arecae semen* (Figure 1). With the continuous improvement and updates to the Chinese Pharmacopoeia, the mycotoxin contamination in traditional Chinese medicines remain a subject of concern for consumers.

The most common detected mycotoxins in traditional Chinese medicines include AFB_1_, AFB_2_, AFG_1_, AFG_2_, ZEN, DON, OTA, T-2 toxin, FB_1_, FB_2_, and PAT [13,14], which are contaminated by the fungal genera *Penicillium*, *Fusarium*, *Aspergillus*, and *Alternaria*. In general, aflatoxins and ochratoxins are primarily produced by the *Aspergillus* species (such as, *A. flavus*, *A. parasiticus*, *A. ochraceus*, *A. niger*, and *A. carbonarius*). *Fusarium* toxins are mainly produced by *Fusarium* spp. (including *F. graminearum*, *F. verticillioides*, *F. equiseti*, *F. culmorum*, and *F. asiaticum*), mainly including ZEN, DON, T-2 toxin, and FBs, among others. Patulin is primarily produced by *Penicillium expansum* [15,16]. Based on the carcinogenicity of mycotoxins, the International Agency for Research on Cancer (IARC) classified mycotoxins into three groups in 1993. AFs were classified as Group 1 (carcinogenic to humans) due to their sufficient evidence of carcinogenicity in humans; OTs and FB_1_ were classified as Group 2B (possibly carcinogenic to humans), for which there is limited but not conclusive evidence of carcinogenicity from human, experimental animal and/or mechanistic data. ZEN, DON, T-2 toxin, and PAT were classified as Group 3 (not classifiable as to its carcinogenicity to humans), for which there is limited and inadequate evidence, or no data in human and experimental animals to classify these [17] (Figure 2).

Group 1 mycotoxins (carcinogenic to humans), which includes aflatoxins (AFs), are secondary metabolites with a dihydrofuran coumarin structure, and are produced by *Aspergillus* spp., particularly *A. flavus* and *A. parasiticus* [4,18,19]. More than 20 kinds of aflatoxins with clearly defined structures have been found, including aflatoxin B_1_, B_2_, G_1_, G_2_, M_1_, M_2_, B_2a_, G_2a_, BM_2a_, and GM_2a_, among others [20]. AFB_1_, AFB_2_, AFG_1_, and AFG_2_ are the main aflatoxins contamination in traditional Chinese medicines, among which, AFB_1_ is the most common and the most toxic aflatoxin. In recent years, there have been many reports about AF contamination in traditional Chinese medicines. Su et al. detected AFB_1_ at a concentration of 4.54 μg/kg in *Astragalus membranaceus* infected by *Aspergillus* and *Penicillium* species [5]. Yang et al. detected AFs (AFB_1_, AFB_2_, AFG_1_, and AFG_2_) in 34 batches of traditional Chinese medicines, AFB_1_ and AFB_2_ were detected in 3 batches of *Semen ziziphi spinosae*, among them, the detection values of AFB_1_ and AFB_2_ were 0.41~1.47 μg/kg and 2.74~20.98 μg/kg, respectively. Yang et al. also reported the presence of AFG_1_ in *Paeoniae radix alba*, with a concentration of 0.45 μg/kg [21]. Wang et al. found that two batches of *Licorice root* samples and one batch of *Fritillary bulbs* tested positive for AFB_1_. The AFB_1_ content in the *Licorice root* was 12.56 μg/kg and 26.11 μg/kg, while the AFB_1_ content in the *Fritillary bulbs* was 10.06 μg/kg, all of which exceeded the maximum limits of 5 μg/kg [22]. High levels of AFB_1_, up to 52.0 μg/kg, were detected in *Platycladi* [23]. A total of 30.8% of *Nutmeg* samples were contaminated with AFB_1_, at levels ranging from 0.73 to 16.31 μg/kg, and 23.1% of the analyzed samples were co-contaminated with at least two different mycotoxins (AFB_1_, AFB_2_, AFG_1_, and AFG_2_) [24]. AFs have been classified as Group 1 mycotoxins (carcinogenic to humans). They are hepatotoxic, mutagenic, teratogenic, carcinogenic and immunosuppressive, posing severe threats to human health [25,26,27,28].

Group 2B mycotoxins (possibly carcinogenic to humans), which include ochratoxins (OTs), are one of the most common naturally occurring mycotoxins produced by *A. ochraceus*, *A. niger*, *A. carbonarius*, *P. viridicatum*, and *P. verrucosum*, as well as other *Penicillium* species [29,30,31]. OTs are isocoumarin toxins, including seven structurally similar compounds, among which OTA, OTB, OTC, and OTD are the main varieties. OTA is the most widely found ochratoxin in traditional Chinese medicines, and it exhibits the highest toxicity within the ochratoxin group [32]. Yang et al. reported that, for 31 of the visibly moldy traditional Chinese medicines and for 26 of the not visibly moldy samples tested, 74.2% of the visibly moldy samples and 7.7% of the not visibly moldy were contaminated with OTA. Specifically, OTA contamination was detected in *Licorice root*, *Mongolian milkvetch root*, *Baikal skullcap root*, *Barley sprout*, *Medicated leaven*, *Common turmeric rhizoma*, *Upland cotton root*, and *Sowthistle tasselflower herb* at different levels in the range of 1.3~84.4 μg/kg, 87.7~158.7 μg/kg, 1.6~1.8 μg/kg, 10.7 μg/kg, 2.4 μg/kg, 10.6 μg/kg, 27.1 μg/kg, and 12.9 μg/kg, respectively [33]. High levels of OTA (92.3 μg/kg) were also detected in tangerine seeds [23]. OTA was detected in the *Massa medicata fermentata*, with the concentration ranging from 1.2 to 5.9 μg/kg [34]. OTA is known to be nephrotoxic, hepatotoxic, immunosuppressive, genotoxic, neurotoxic, carcinogenic, teratogenic, and embryotoxic, posing severe risks to human health [35,36,37,38].

Fumonisins (FBs) are a group of secondary metabolites produced by *Fusarium* spp., such as *F. verticillioides*, *F. proliferatum*, *F. oxysporum*, and other related species. They are classified into four categories (A, B, C, and P), with the B series being the most significant and widely studied [39,40]; among these, FB_1_ and FB_2_ are the most common toxins. FB_1_ is the strongest level toxicity, with heat-stable and water-soluble properties [19]. Li et al. detected FB_1_ and FB_2_ in *Eupolyphaga steleophaga* and Chinese medicinal preparations containing *Eupolyphaga steleophaga*; among these samples, the content of FB_1_ and FB_2_ was 13.51 μg/kg and 6.31 μg/kg, respectively. In Chinese medicinal preparations containing *Eupolyphaga steleophaga*, the detection values of FB_1_ and FB_2_ ranged from 2.38 μg/kg to 17.87 μg/kg and from 3.01 μg/kg to 7.89 μg/kg, respectively [41]. Wang et al. established a determination method for 26 kinds of mycotoxins with UPLC-MS/MS in *Notoginseng radix et rhizoma*, and one batch sample detected FB_2_, with a detection value of 1.2 μg/kg [42]. Among 20 batches of *Ophiopogonis radix* samples, FB_2_ was detected in 14 samples, with the levels ranging from 4.99 to 378.99 μg/kg, and a positive detection rate of 70% [43]. FBs mainly damage the heart, lungs, liver, kidneys, and other organs of animals. Additionally, the exposure to FBs has been linked to human esophageal cancer and neural tube defects [39,44].

Group 3 mycotoxins (not classifiable as to its carcinogenicity to humans) include ZEN, DON, T-2 toxin, and PAT. ZEN is an estrogenic mycotoxin produced by *Fusarium* spp. (primarily including *F. graminearum*, *F. asiaticum*, *F. equiseti*, and *F. culmorum*) under specific environmental conditions, such as high humidity and moderate temperatures [45]. ZEN is also a non-steroidal estrogenic toxin [46], which often co-occurs with DON and AFs [47]. The natural occurrence of ZEN has been reported in many countries [48,49,50]. ZEN contamination was detected in 107 tested *Coix seed* samples, with 55.6% of them exceeding the Chinese regulatory standards [51]. The detection rate of ZEN in *Coix seed* samples is relatively high, with the content of ZEN ranging from 0.343 μg/kg to 129.767 μg/kg [52]. ZEN contamination levels in the *Coicis semen* samples ranged from 2.77 μg/kg to 1076.18 μg/kg, with a detection rate of 84% and an over-standard rate of 6% [53]. Xu et al. detected ZEN in *Quisqualis indica L*., with a detection rate of 1.52% [54]. Zhang et al. reported the levels of ZEN in *Massa medicata fermentata* ranging from 22.0 to 557.0 μg/kg [34]. ZEN mainly damages human endocrine and reproductive functions, leading to adverse effects, such as infertility, miscarriage, and other phenomena [55].

DON is one kind of trichothecene mycotoxin produced by the *Fusarium* species (mainly including *F. graminearum*, *F. culmorum*, *F. pseudograminearum*, and *F. sporotrichioides*) during the growth of agricultural crops [38]. It is the most common mycotoxin contaminating traditional Chinese medicines. DON is a representative of the type-B trichothecenes [56,57], and its contamination is widespread. In *Coicis semen* samples, DON contamination levels ranged from 1.54 to 360.12 μg/kg, with a detection rate of 36% [53]. Ten mycotoxins were detected in 483 traditional Chinese medicines, with a detection rate of 4.8%, and an average content of 52.9 μg/kg in positive samples [58]. DON mainly causes vomiting, diarrhea, gastrointestinal discomfort, headache, nausea, and other symptoms in humans and animals. In severe cases, DON can damage the hematopoietic system and even lead to death, and can often have synergistic effects with other mycotoxins [59,60].

T-2 toxin is a type A trichothecene mycotoxin primarily produced by certain *Fusarium* species, notably *F. sporotrichioides* and *F. langsethiae* under specific environmental conditions. It is a representative of the type-A trichothecene that is one of the most toxic mycotoxins among the type-A trichothecene. Li et al. reported T-2 contaminations in *Polygonum multiflorum* and *Panax notoginseng*, and the content level was 1.93 μg/kg [61]. Zhao et al. detected 10 mycotoxins in 30 batches of traditional Chinese medicines. Among them, the positive detection rates were 80%, 60% and 70%, and the average concentration levels were 1.19 ± 0.54 μg/kg, 0.86 ± 0.61 μg/kg, and 1.00 ± 0.45 μg/kg for *Polygalae radix*, *Coicis semen*, and *Eupolyphaga steleophaga*, respectively, which were all less than 2.0 μg/kg [62]. Although the contamination levels of T-2 toxin reported in traditional Chinese medicines were low, its toxicity should be paid attention to. T-2 toxin can directly stimulate the skin and mucosa, causing inflammation, and can also damage bone marrow hematopoietic tissue, enhance capillary permeability, and cause incomplete blood coagulation, resulting in visceral bleeding [15,63]. In addition, T-2 toxin can cause pathological changes in liver tissue and damage to the immune system [64].

PAT, also known as patulin, is a polyacetal lactone compound that is a secondary metabolite generated by *Penicillium*, *Aspergillus*, *Byssochlomys*, and *Paecilomyces* species under favorable conditions; it is usually found in pome fruits and their derivative products [65]. Zhang et al. found the blue mold tissues of Lanzhou Lily to be contaminated with patulin, produced by *P. gladioli* and *P. polonicum*, with a content of 2.38 μg/mL and 1.51 μg/mL, respectively [66]. PAT was detected in fresh *Codonopsis pilosula* infected by *P. expansum*, and HT-2 toxin and 15-acetyl-deoxynivalenol were also found in fresh *Codonopsis pilosula* infected by *F. acuminatum*, while DON, HT-2 toxin, and 15-acetyl-deoxynivalenol were detected in fresh *Codonopsis pilosula* infected by *Trichothecium roseum* [67]. PAT, sterigmatocystin, and 15-acetyl-deoxynivalenol were detected in fresh *Angelica sinensis* inoculated with *P. polonicum*, *Aspergillus versicolor*, and *F. solani*, respectively [68]. All the above results indicated that PAT existed widely in traditional Chinese medicines. PAT has broad toxicity, and exposure to PAT can cause a variety of symptoms in humans and animals, including nausea, vomiting, convulsions, and coma [69]. Furthermore, PAT has cytotoxicity, teratogenicity, carcinogenicity, and mutagenicity toxicity, as well as immunotoxicity at high doses [70,71,72].

## 3. Detection Methods of Mycotoxins in Traditional Chinese Medicines

### 3.1. Detection Methods and Limited Standards of Mycotoxins in the Chinese Pharmacopoeia

The 2010 edition of the Chinese Pharmacopoeia stipulates the AF determination method to be high performance liquid chromatography (HPLC), including high performance liquid chromatography–iodine derivatization and high performance liquid chromatography–photochemical derivatization. The 2015 edition of the AF determination method adds high-performance liquid chromatography tandem mass spectrometry (HPLC-MS/MS). The 2020 edition of the AF determination method adds enzyme-linked immunosorbent assay (ELISA), as well as HPLC and HPLC-MS/MS for the determination of OTA, ZEN, and DON, HPLC-MS/MS for PAT, and high performance liquid chromatography with fluorescence detection (HPLC-FLD) for the determination of ZEN. The appendix includes HPLC-MS/MS for the determination of multiple mycotoxins, covering a total of seven categories and 11 types of toxins, namely AFB_1_, AFB_2_, AFG_1_, AFG_2_, OTA, DON, ZEN, PAT, FB_1_, FB_2_, and T-2 toxin.

The 2020 edition of the Chinese Pharmacopoeia stipulates a permissible limit of 5 μg/kg for AFB_1_ and 10 μg/kg for the total AFs (the sum of AFB_1_, AFB_2_, AFG_1_, and AFG_2_) in 24 types of traditional Chinese medicines. It also adds a permissible limit of 500 μg/kg for ZEN in *Coicis semen* for the first time (Table 1). There are no detection methods developed for other mycotoxins, and there is no clear limit standard.

### 3.2. The Process of Common Detection Methods for Mycotoxins

It is well known that traditional Chinese medicines have a complex matrix, which makes it extremely difficult to determine the contamination of mycotoxins. However, with the rapid and continuous development of detection technology and the diversification of detection techniques, it is possible to meet the needs of the detection of mycotoxins in complex matrices. In recent years, the major detection methods for common mycotoxins in traditional Chinese medicines mainly included HPLC, HPLC-MS/MS, rapid detection technology, and field real-time monitoring technology. A variety of pre-treatment methods have currently been implemented. The most commonly used methods for mycotoxin treatment are solid phase extraction (SPE), immune-affinity columns (IAC), multifunctional cleanup columns, column chromatography, and QuEChERS. A flow diagram of common steps for mycotoxin analysis in traditional Chinese medicines is presented in Figure 3. These technologies have their own advantages and disadvantages, and should be selected according to different application scenarios to provide help for the detection, analysis, and quality safety control of mycotoxins in traditional Chinese medicines. The contamination status of mycotoxins detected by these instrumental methods in traditional Chinese medicines are shown in Table 2. 

#### 3.2.1. Liquid Chromatography

High performance liquid chromatography (HPLC) is based on the different retention time of different components in the chromatography column to be separated effectively, which places higher requirements on the sample, as most of the pretreatment requires immunoaffinity column to clean up and derivatization determination, including pre-column derivatizations [73], post-column derivatizations [21,74], photochemical derivatizations, and electrochemical derivatizations [75]. The pretreatment steps are complex, the detection cost is high, often only used to determine a single mycotoxin or a group of chemically related mycotoxins, and not suitable for the screening of large-batch samples. Yang et al. established HPLC methods combined with post-column photochemical derivatization after immunoaffinity column purification to determine aflatoxin B_1_, B_2_, G_1_, and G_2_ in 34 batches of traditional Chinese medicines. Among them, AFs were detected in three batches of *Semen ziziphi spinosae*, and AFB_1_ in one batch exceeded the national standard limit (5 μg/kg) [21]. Yin et al. detected the content of four aflatoxins (AFB_1_, AFB_2_, AFG_1_, AFG_2_) in 21 kinds of traditional Chinese medicines by high performance liquid chromatography–post column iodine derivatization. All four batches of *Platycladi semen* and six batches of *Coicis semen* were contaminated by aflatoxins, with a positive detection rate of 100%. Moreover, the aflatoxin content in *Platycladi semen* and *Coicis semen* exceeded the legal standard limit, with an unqualified rate of 75% and 33%, respectively [76]. Kong et al. simultaneously determined the content of aflatoxins (AFB_1_, AFB_2_, AFG_1_, AFG_2_) and OTA in 13 batches of *Myristicae semen* using ultrasound-assisted solid–liquid extraction and immunoaffinity column clean-up combined with HPLC and on-line post-column photochemical derivatization–fluorescence detection (USLE-IAC-HPLC-PCD-FLD), and found that four batches of *Myristicae semen* were detected with contamination by AFs and one with OTA, the detection rate of AFB_1_ was 30.8%, with the content levels of 0.73–16.31 μg/kg. Furthermore, at least two different mycotoxins co-occurred in three samples [24]. Yang et al. used immunoaffinity column purification combined with HPLC-fluorescence detection (HPLC-FLD) to determine OTA in 57 batches of traditional Chinese medicines, such as *Licorice*, *Astragalus*, *Scutellaria baicalensis*, *Barley*, and *Angelica*, among others. The results showed that both moldy and non-moldy samples of traditional Chinese medicines may be contaminated by OTA, and the levels of contamination are different. A total of 23 of the visibly moldy samples and 2 of the not visibly moldy were contaminated with OTA levels in the range of 1.2–158.7 μg/kg and 2.5–5.6 μg/kg, respectively [33]. Yang et al. established a method of immunoaffinity column purification combined with HPLC-FLD to detect aflatoxins B_1_, B_2_, G_1_, and G_2_ in traditional Chinese medicines, such as *Herba ephedra*, *Platycladi semen*, *Corydalis rhizoma*, and *Coicis semen*, among others. The aflatoxin detection rate was 15.78%, with the content of the total aflatoxins ranging from 2.90 μg/kg to 32.18 μg/kg [77]. Zheng et al. determined AFs in 56 batches of 11 kinds of traditional Chinese medicines by HPLC, and found that AF contamination was more severe in *Bombyx batryticatus*, *Ziziphi spinosae semen*, *Coicis semen*, and *Sterculiae lychnophorae semen*; moreover, their group also suggested that samples with aflatoxins did not present apparent moldy symptom [78]. A total of 75 batches of traditional Chinese medicines were analyzed by the established method of immunoaffinity column purification–high performance liquid chromatography fluorescence detector (IAC-HPLC-FLD), the results showed that the positive detection rates of aflatoxins in *Platycladi seeds*, *Coix seed*, and *Cassia* were 77%, 29%, and 7%, respectively [79]. Wang et al. found the mycotoxin contamination level of AFB_1_, AFB_2_, and ZEN in *Coix seed* was different under different storage conditions (such as cryogenic seal, high temperature and humidity, aeration, and drying) by HPLC-FLD [80]. In brief, the contamination of mycotoxins in traditional Chinese medicines is related to the sample types or species.

#### 3.2.2. Liquid Chromatography–Tandem Mass Spectrometry (LC-MS/MS)

High performance liquid chromatography–tandem mass spectrometry (HPLC-MS/MS) has become the mainstream method for the qualitative and quantitative analysis of mycotoxins and their metabolites in traditional Chinese medicines, due to its high separation performance in chromatography, its specificity and high sensitivity in mass spectrometry, and its simultaneous determination of multiple mycotoxins, which greatly improves the detection efficiency of samples. Moreover, the method of LC-MS/MS does not require too high a purity for test samples, and is suitable for large-scale sample screening and simultaneous determination of multi-component samples, which has certain advantages over samples with complex matrices especially.

Traditional Chinese medicines often contain various components, such as lipids, proteins, starch, tannins, and volatile oils, among other components. HPLC-MS/MS is more suitable for traditional Chinese medicines (rich in lipids, proteins, starch, and tannins) determination. Wu et al. employed HPLC-MS/MS to determine the traditional Chinese medicines that are rich in protein and lipid substrates, and found that the mycotoxin detection rate was higher in these traditional Chinese medicines, while the detection rate of mycotoxin was lower in traditional Chinese medicines containing high starch, volatile oils, and cellulose. It can be seen that the production of mycotoxins has a certain correlation with the composition of the matrix [1]. The complex matrix components can cause interferences to the detection; nevertheless, the advantages of LC-MS/MS can effectively reduce the matrix interference. Currently, numerous studies have reported on the simultaneous detection of multiple mycotoxins. It can be seen from Table 1 that there are multiple mycotoxin co-occurrences in a single kind of traditional Chinese medicines, such as *Myristicae semen* [81,82], *Eupolyphaga steleophaga* [41], *Angelica sinensis* [83], *Persicae semen* [84], *Coicis semen* [53], *Astragali radix* [85], *Notoginseng radix et rhizoma* [42,54], and *Platycladi semen* [54]; whereas LC-MS/MS can screen multiple mycotoxins simultaneously, and will not be limited by the types of mycotoxin. Zhao et al. established a determination method for 10 mycotoxins using a multifunctional immune affinity column (multi-IAC) combined with HPLC-MS/MS for traditional Chinese medicine materials, and investigated a total of 30 samples (such as *Polygalae radix*, *Coicis semen*, and *Eupolyphaga steleophaga*) with mycotoxin contamination and co-occurrence. The results showed that all the samples in this study contained more than five mycotoxins, and AFB_1_–AFs, AFB_1_–DON, AFB_1_–FBs, and AFB_1_–T-2 are the most observed co-occurrences; a synergistic toxicity was also observed between AFB_1_–OTA [62]. Li et al. established an analytical method for the determination of 16 mycotoxins in traditional Chinese medicine materials by isotope labeling–ultra performance liquid chromatography tandem mass spectrometry (UPLC-MS/MS), and found that 10 mycotoxins were detected in the 483 batches of samples, and ZEN has the highest detection rate, with an average content of positive samples of 71.2 μg/kg; among them, 3.11% of the samples exceeded the reference limit specified in the national standards [58]. Li et al. developed and established a UPLC-ESI-MS/MS method to simultaneously determine 12 mycotoxins (AFB_1_, B_2_, G_1_, G_2_, OTA, OTB, FB_1_, FB_2_, T-2 toxin, HT-2 toxin, DON, and ZEN) in *Polygoni multiflori radix*, and found that 8 kinds of mycotoxins, including AFB_1_, AFG_2_, FB_1_, FB_2_, OTA, OTB, T-2, and HT-2, were detected; their contents ranged approximately from 0.51 to 1643.2 μg/kg. Among them, AFB_1_ in one batch of sample, with the content of 6.8 μg/kg, was beyond its limit standard [61]. Qin et al. determined 10 mycotoxins in 10 batches of *Dichondra micrantha UID* by UPLC-MS/MS, and the positive rate of the samples was 70% and 6 mycotoxins were detected; at the same time, the mycotoxins had a mixed contamination [86]. The types of mycotoxins contaminated in different traditional Chinese medicines are different [43,87,88]. The advantages of LC-MS/MS detection technology can meet the needs of the simultaneous determination of different kinds of toxins, and it is also a powerful technical means for confirming the mycotoxins in traditional Chinese medicines.

#### 3.2.3. Rapid Detection Technology

To monitor and control the contamination by mycotoxins, rapid detection technology has become the research focus due to its relative simplicity, convenience, accuracy, and efficiency. Rapid detection techniques mainly include enzyme-linked immunosorbent assay (ELISA) and immunochromatography.

Enzyme-linked immunosorbent assay (ELISA) is a detection technique based on antigen and antibody specific binding reactions, which opens up a new field of rapid mycotoxin analysis. This technology has the advantages of simple pretreatment, rapid detection speed, strong specificity, high sensitivity, and convenience. ELISA can be used for the preliminary screening of mycotoxins in large numbers of samples, and is suitable for on-site rapid detection, both of which are commonly employed in the field. However, the method has some drawbacks, including numerous preparation steps, washing frequency, and long diffusion times required for antigen–antibody binding [89], thus it is time-consuming. The results have poor reproducibility, poor enzyme stability, and limited application in fast detection. Occasionally, it also produces false positive results [90]. Yang et al. determined AFB_1_ in seven *Dendrobium candidum* samples with known concentrations by the ELISA method, and found that the results of AFB_1_ content were basically consistent with those of HPLC [91]. Liu et al. determined the total AFs in *Ziziphi spinosae semen*, *Persicae semen*, *Hordei fructus germinatus*, *Polygalae radix*, *Cassiae semen*, and *Platycladi semen* by ELISA, and the results were basically consistent with those of HPLC [92]. Currently, using the ELISA method can specifically identify the contents of AFB_1_ and the total AFs [93,94]. ELISA is commonly used for the detection of AFs, OTA, ZEN, and DON [90]. Among the established ELISA method for detecting mycotoxins in traditional Chinese medicines, AFB_1_ is the most mature and widely applied. The 2020 edition of the Chinese Pharmacopoeia first included the ELISA method, providing a technical option for the effective control of mycotoxins in traditional Chinese medicine materials.

Immunochromatographic assay (ICA) is a rapid immunoassay method that adopts colloidal gold or quantum dots as a labeled probe to detect multiple components by the specific binding reaction of antigen and antibody. IAC has the characteristics of rapid, simple, low cost, high sensitivity, and strong specificity, and is a research focus in the field of rapid detection. Huo et al. established a simultaneous and rapid detection method of mycotoxins (such as AFB_1_, OTA, and ZEN) in *Ginseng*, *Astragalus*, and *Angelica* samples by fluorescent nanoparticle-based immunochromatography, and found that the coincidence rate of the results between the actual sample and the HPLC method was consistent [94]. Hou et al. employed colloidal gold immunochromatography to achieve the one-step rapid detection of FB_1_, DON, and ZEN. The results were consistent with those of LC-MS/MS, which can verify the feasibility of this method [95]. However, the immunochromatographic detection technology also has some problems, such as poor stability, difficulty in long-term storage, and inaccurate quantification. Zhang et al. used IAC technology to capture and concentrate OTA in medicinal samples, and then determined it using ELISA technology. The two technologies were used in synergy for detection, and the sensitivity was significantly improved [96]. On one hand, by combining IAC with ELISA technology, it can improve the sensitivity of the method; on the other hand, it does not require that sensitivity to employ large-scale equipment to operate the detection. This method not only has good applicability, but is conducive to the preliminary screening of large quantities of samples, which can provide new ideas for the development of detection technology in the future.

#### 3.2.4. Field Real-Time Monitoring Technology

Field real-time monitoring technologies include the spectroscopy method and sensor technology. The spectroscopic method is widely applied in the field of food for the determination of mycotoxins due to its unique advantages of no sample pretreatment, good reproducibility, and rapid detection. However, its application is rare in traditional Chinese medicines. The sensor method mainly includes immunosensors, aptamer sensors, enzyme sensors, and other sensors used for mycotoxin detection, all of which have the characteristics of rapid detection, simplicity and convenience, good specificity, high sensitivity, and low cost. They are suitable for the trace detection of small molecule substances and are an effective means for mycotoxin identification [97,98]. Fan et al. designed electrochemical biosensors for AFB_1_ and OTA detection in Chinese herbal medicines [99]. Although the detection limit and recovery rate can meet the detection requirements of mycotoxins in Chinese herbal medicines, nevertheless, most sensors can only detect one toxin, and they are unable to meet the actual requirements of multiple mycotoxins at the same time. The rapid, convenient, low-cost, and high specificity of rapid detection technology has a driving effect on the preliminary screening of samples. Therefore, there are still some challenges in the practical application of these technologies.

**Table 2 jof-11-00411-t002:** Detection technology and contamination status of mycotoxins in traditional Chinese medicines.

Detection Method	Sample	Pretreatment Technology	The Number of Toxins Detected	Toxin Species Detected	Ref.
LC (including fluorescence detector with post column photochemical derivatization, or post-column iodine derivatization)	*Ziziphi spinosae semen*, *Citri reticulatae pericarpium*, *Paeoniae radix alba*, *Angelica sinensis radix*, *Ophiopogonis radix*, *Schisandra chinensis fructus*, *Isatidis radix*, *Astragali radix*, *Bupleuri radix*, *Salviae miltiorrhizae radix et rhizoma*	IAC	4	AFB_1_	[21]
	*Sterculiae lychnophorae semen*, *Ziziphi spinosae semen*, *Jujubae fructus*, *Scorpio*, *Sojae semen praeparatum*, *Arctii fructus*, *Euphorbiae semen*, *Platycladi semen*, *Coicis semen*, *Persicae semen*, *Citri reticulatae pericarpium*	IAC	4	AFB_1_, AFB_2_, AFG_1_, AFG_2_	[76]
	*Myristicae semen*	Ultrasound assisted solid–liquid extraction and IAC	5	AFB_1_, AFB_2_, AFG_1_, AFG_2_, OTA	[24]
	*Astragali radix*, *Scutellaria radix*, *Angelica sinensis radix*, *Glycyrrhizae radix et rhizoma*, *Morindae officinalis radix*, *Atractylodis rhizoma*, *Salviae miltiorrhizae radix et rhizoma*, *Panacis quinquefolii radix*	IAC	9	OTA	[33]
	*Corydalis rhizoma*, *Armeniacae semen amarum*, *Ephedrae herba*, *Platycladi semen*	IAC	4	Total AFs	[77]
	*Jujubae fructus*, *Angelica sinensis radix*, *Astragali radix*, *Bombyx batryticatus*, *Sterculiae lychnophorae semen*, *Ginseng radix et rhizoma*, *Notoginseng radix et rhizoma*, *Persicae semen*, *Coicis semen*, *Citri reticulatae pericarpium*, *Ziziphi spinosae semen*	IAC	4	Total AFs	[78]
	*Platycladi semen*, *Coicis semen*, *Cassiae semen*, *Codonopsis radix*	IAC	4	AFB_1_, Total AFs	[79]
	*Coicis semen*	IAC	7	AFB_1_, AFB_2_, AFG_1_, AFG_2_, ZOL, ZEN	[80]
HPLC-MS/MS	*Myristicae semen*	IAC	12	OTA, AFB_1_, AFB_2_	[81]
	*Myristicae semen*	QuEChERS	21	AFB_1_, AFB_2_, AFG_1_, AFG_2_, AFM_1_	[82]
	*Eupolyphaga steleophaga*	QuEChERS	9	FB_1_, FB_2_	[19]
	*Angelica sinensis radix*	SPE	9	AFB_1_, AFB_2_, AFG_1_	[81]
	*Persicae semen*	QuEChERS	10	AFB_1_	[82]
	*Coicis semen*	/	14	ZEN, AFB_1_, AFB_2_, DON, ST	[50]
	*Astragali radix*, *Coicis semen*, *Eupolyphaga steleophaga*	SLE-SPE, QuEChERS	21	AFB_1_, AFB_2_, OTA, OTB, PA	[85]
	*Notoginseng radix et rhizoma*	SPE	26	FB_1_	[42]
	*Quisqualis fructus*	SPE	22	AFB_1_, AFB_2_, AFM_1_, OTA, OTB, ZEN	[54]
	*Coicis semen*, *Polygalae radix*, *Eupolyphaga steleophaga*, *Polygalae Radix*, *Eupolyphaga Steleophaga*	multi-IAC	10	AFB_1_–AFs, AFB_1_–FBs, AFB_1_–DON, AFB_1_–T-2, AFB_1_–OTA	[85]
	*Coicis semen*, *Galli gigerii endothelium corneum*, *Hordei fructus germinatus*, *Persicae semen*, *Dioscoreae rhizoma*, *Poria*	multifunction clean-up columns	16	AFB_1_, AFB_2_, AFG_1_, ZEN, DON, OTA, NIV, FB_1_, FB_2_, FB_3_	[58]
	*Polygoni multiflori radix*	Modified QuEChERS	12	AFB1, AFG1, T-2, HT-2, OTA, OTB, FB_1_, FB_2_	[61]
	*Dichondra micrantha* Urb.	Modified QuEChERS	10	AFB_1_, AFG_1_, ZEN, OTA, DON, FB_1_	[86]
	*Glehniae radix*, *Astragali radix*, *Codonopsis radix*	mPFC- QuEChERS	16	OTA, ST, 15-ACE	[87]
	*Polygonati rhizoma*, *Ophiopogonis radix*	SPE	10	FB_2_, ZEN	[43]
	*Lonicerae flos*, *Puerariae lobatae radix*, *Hippophae fructus*	Accelerated solvent extraction (ASE)-QuEChERS	16	AFB_1_, AFG_2_, OTA, FB_1_	[88]
Immunochromatographic detection technique	*Dendrobium candidum*	ELISA	1	AFB_1_	[91]
	*Ziziphi spinosae semen*, *Persicae semen*, *Hordei fructus germinatus*, *Polygalae radix*, *Cassiae semen*, *Platycladi semen*	ELISA	4	Total AFs	[92]
	*Ground beetle*, *cockroach*, *Silkworm*, *Earthworm*	ELISA	1	AFB_1_	[100]
	*Jujubae fructus*, *Nelumbinis semen*, *Coicis semen*, *Poria*, *Lilii bulbus*, *Euryales semen*, *Sojae semen nigrum*, *Pseudostellariae radix*, *Cinnamomi cortex*, *Phragmitis rhizoma*, *Dioscoreae rhizoma*, *Codonopsis radix*, *Crataegi fructus*, *Vignae semen*, *Prunellae spica*	ELISA	4	AFB_1_, Total AFs	[91]
	*Ginseng radix et rhizoma*, *Astragali radix*, *Angelica sinensis radix*	Lateral flow immunochromatography, LFIC	3	ZEN, AFB_1_, OTA	[94]
	*Lotus seed*	ELISA	1	AFB_1_	[101]
	*Fallopia multiflora*, *Codonopsis pilosula*, *Apricot kernel*, *Zingiber officinale*, *Wild jujube seed*, *Malt*, *Cassia obtusifolia*, *Red lotus seed*	immunochromatographic test strip (ICS)	6	AFB_1_, ZEN, T-2	[102]
	*Astragali radix*, *Scutellariae radix*, *Glycyrrhizae radix et rhizoma*, *Atractylodis macrocephalae rhizoma*, *Bupleuri radix*, *Lonicerae japonicae flos*, *Phellodendri chinensis cortex*, *Chuanxiong rhizoma*, *Isatidis radix*, *Pinelliae rhizoma*, *Notoginseng radix et rhizoma*, *Polygonati rhizoma*, *polygoni multiflori radix*	IAC and ELISA	1	OTA	[96]
Field real-time monitoring technology	*Coicis semen*, *Jujubae fructus*, *Cassiae semen*	electrochemical biosensor	1	AFB_1_	[98]
	*Nelumbinis semen*, *Panacis quinquefolii radix*	electrochemical biosensor	1	OTA	[99]

## 4. Comprehensive Prevention and Control Strategies for Mycotoxins in Traditional Chinese Medicines

Traditional Chinese medicines can be contaminated by mycotoxins under unfavorable conditions of growth, processing, storage, and transportation. The production of mycotoxins can be avoided by taking prevention and control strategies during the whole industry chain for the processing of traditional Chinese medicines. Thus, some strategies for controlling the mycotoxin contamination from the industrial chain need to be urgently proposed. Firstly, it is necessary to reduce the possibility of mycotoxin contamination from the source through to selecting the resistant variety of traditional Chinese medicines, and to standardize field operation technology and cultivation management procedures, to prevent traditional Chinese medicines from being infected by pathogenic fungi during planting.

To prevent and control the production of mycotoxins in traditional Chinese medicines throughout the entire process, the following steps should be taken. After harvesting, it is extremely crucial to keep traditional Chinese medicines dried in a timely manner, and to control the temperature and humidity to reasonable levels during stacking. At the same time, it is also necessary to standardize the drying, sterilization, and other operating methods during the processing of traditional Chinese medicines. Moreover, it is essential to strictly control the temperature, humidity, and the moisture content of products in storage and transportation. Furthermore, it is essential to select suitable packaging materials, and to establish processing and storage standards for traditional Chinese medicines.

In order to improve the quality and safety of traditional Chinese medicines, it is also necessary to strengthen the awareness of the relevant personnel in the prevention and control of mycotoxin contamination through training, which can improve the concept of standardized planting, the technical level and monitoring ability of planting personnel, as well as to collaborate with professionals from scientific research institutions to explore the prevention and control measures of mycotoxins.

## 5. Regulatory Recommendations for Mycotoxins Contamination in Traditional Chinese Medicines

### 5.1. Improve the Detection Standards

Mycotoxin contamination is one of the important risk factors affecting the quality and safety of traditional Chinese medicines. How to quickly, accurately, and efficiently detect mycotoxins in traditional Chinese medicines in large quantities has been the focus of researchers. Although the 2020 edition of the Chinese Pharmacopoeia has listed some detection methods for mycotoxins, it is still lacking for some research on mycotoxins in traditional Chinese medicines and their preparations. At present, an instrumental method is the main detection and analysis technology included in the Chinese Pharmacopoeia, which has some issues, such as high experimental costs, strong operability, high technical requirements for inspectors, and long inspection time. However, it is reliable in terms of the accuracy and stability of the qualitative and quantitative analysis. The field rapid detection and online monitoring technologies have great application prospects because of their characteristics of high sensitivity, good repeatability, and low cost. However, there are some challenges in practical applications. Currently, the rapid detection method has not been included in the Chinese Pharmacopoeia. Traditional Chinese medicines have complex matrix components, of which the chemical structure of some mycotoxins is similar. During the detection process, it is not only affected by matrix interference, but the chemical structures and chromatographic behaviors of most mycotoxins are similar, making it challenging for the efficient analysis of multiple mycotoxin residues in traditional Chinese medicines. It is urgent to establish an efficient, accurate, and reliable monitoring system for multiple mycotoxins, and it is recommended to accelerate the formulation and improvement of relevant testing method standards.

### 5.2. Supplement Limit Regulations

At present, the limits of mycotoxins set for traditional Chinese medicines in the pharmacopoeia of many countries are relatively few, which cannot meet actual needs. The 2020 edition of the Chinese Pharmacopoeia only stipulates the limits for AFs and ZEN, and only includes 24 types of traditional Chinese medicines, which are far fewer than the actual number of medicinal materials found to be contaminated with mycotoxins. At the same time, the co-occurrence of multiple mycotoxins has not been given enough attention, and no limits have been established. The existing limit standards cannot meet the practical needs, making it difficult to evaluate and judge some test results for which there are no regulatory restriction standards on other traditional Chinese medicines susceptible to mycotoxins. Therefore, supervision work is also difficult to implement. It is suggested that temporary limit regulations should be set for traditional Chinese medicines susceptible to mycotoxin contamination, and for the mycotoxins frequently detected in traditional Chinese medicines. Afterwards, it is necessary to gradually solve the relatively lagging problem in the formulation of limit standards, and provide a basis for supervision and enforcement.

### 5.3. Strengthen Risk Monitoring Programs

At present, the monitoring data for mycotoxins in China are relatively scattered, and lack systematic monitoring. In addition, a database resource has yet to be formed. The monitoring of mycotoxin contamination levels in Chinese herbal medicine is relatively insufficient, and the role of safety supervision is limited. It is necessary to develop a reasonable risk monitoring plan for mycotoxins in traditional Chinese medicines, and to carry out routine monitoring in a directional manner. It is essential to establish a monitoring system and conduct a risk assessment, to identify high-risk varieties and mycotoxin factors, and to enhance risk warnings.

## 6. Conclusions and Future Perspectives

Mycotoxin contamination in traditional Chinese medicines is extremely universal and widespread, and poses severe health risks due to the multifaceted toxicity of these fungal metabolites. The most common mycotoxins in traditional Chinese medicines include aflatoxins, ochratoxins, zearalenone, deoxynivalenol, fumonisins and patulin, and T-2 toxin, which can affect the human exocrine and endocrine systems and suppress immune function, thus causing teratogenicity, carcinogenicity, and mutagenicity. Current detection methods for mycotoxins, including HPLC, HPLC-MS/MS, rapid detection technology, and field real-time monitoring technology, have played a crucial role in mycotoxin surveillance. However, these methods remain limited by high costs, variable sensitivity, false positives, or impractical field applicability. Thus, a rapid, accurate, highly-sensitive, cost-effective, and deployable, in a real-world setting, detection method for mycotoxins in traditional Chinese medicines still urgently needs to be developed. Moreover, it is essential to prevent and control the mycotoxin contamination from the source. Therefore, some suggestions should be put forward, such as to formulate uniform regulations for preharvest and postharvest, develop rapid and synchronous detection methods, increase legal limit regulations for common varieties of traditional Chinese medicines, as well as to strengthen public awareness of prevention, control, and management.

## Figures and Tables

**Figure 1 jof-11-00411-f001:**
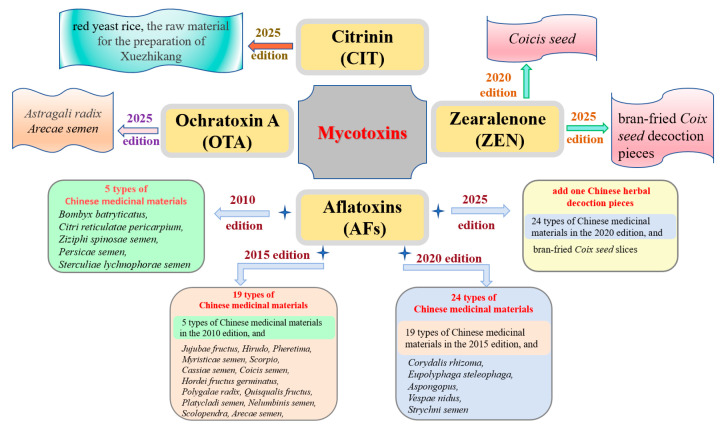
The varieties of traditional Chinese medicines tested for mycotoxins in the 2010 to 2025 editions of the Chinese Pharmacopoeia.

**Figure 2 jof-11-00411-f002:**
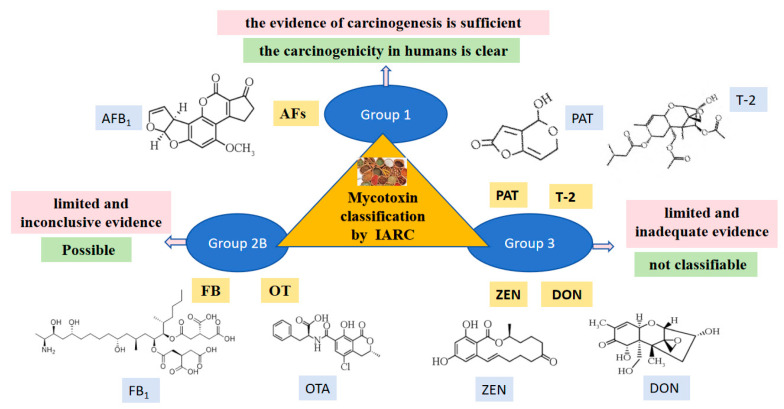
Classification and carcinogenicity of common mycotoxins in the Chinese medicinal material.

**Figure 3 jof-11-00411-f003:**
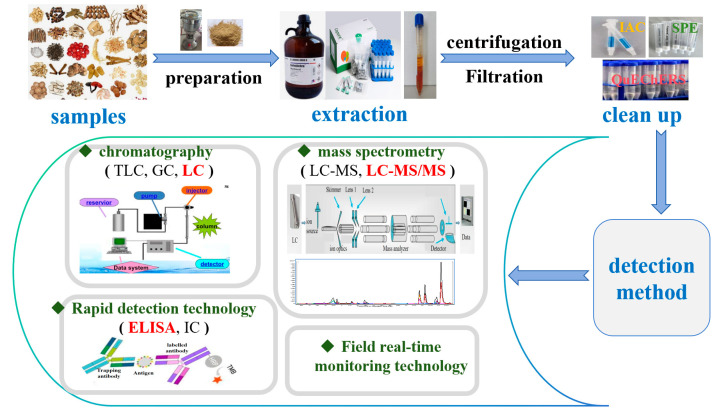
Flow diagram of common steps in mycotoxin analysis.

**Table 1 jof-11-00411-t001:** The limit values of mycotoxins in the 2020 edition of the Chinese Pharmacopoeia.

Traditional Chinese Medicines	The Limit Value of Mycotoxins (μg/kg)		
	Total AFs	AFB_1_	ZEN
*Bombyx batryticatus*, *Citri reticulatae pericarpium*, *Ziziphi spinosae semen*, *Persicae semen*, *Hirudo*, *Sterculiae lychnophorae semen*, *Jujubae fructus*, *Pheretima*, *Myristicae semen*, *Cassiae semen*, *Hordei fructus germinatus*, *Quisqualis fructus*, *Polygalae radix*, *Platycladi semen*, *Scorpio*, *Nelumbinis semen*, *Scolopendra*, *Arecae semen*, *Coicis semen*, *Corydalis rhizoma*, *Aspongopus*, *Eupolyphaga steleophaga*, *Vespae nidus*, and *Strychni semen*.	10	5	/
*Coicis semen*	/	/	500

## Data Availability

The original contributions presented in this study are included in the article. Further inquiries can be directed to the corresponding author.

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
