# Peer review of "Research Progress and Management Strategies for the Common Mycotoxin Contamination of Traditional Chinese Medicines"

_jof, 2025, doi:10.3390/jof11060411_

Round 1

Reviewer 1 Report (Previous Reviewer 3)

Overall, the authors' responses are satisfactory, and we appreciate their taking the time to review the taxonomy of mycotoxin-producing fungal species. The manuscript has been adequately revised. However, some errors should be corrected before publication (see detail comment)

Line 85: I suggest the inclusion of Fusarium asiaticum as a mycotoxin-producing species.

Line 92: Group 3 of IARC means Not classifiable as to its carcinogenicity to humans (it is not correct to say ‘due to their suspected carcinogenicity in humans’).

Line 119: Please do not use the term Group 2B carcinogens as they are not; the correct term is Group 2B Possibly carcinogenic to humans.

Line 120: The species Penicillium viridicatum is not ochratoxin producer and must be deleted from the list. The attribution of ochratoxin A to the species P. viridicatum is based on an initial identification error that has already been clarified for some time. Misidentifying Penicillium species is common due to a lack of a complete sequence database and outdated species lists. This can lead to incorrect cultures and questionable interpretations of results.

Line 153: Please do not use the term Group 3 carcinogen as they are not; Group 3 means Not classifiable as to its carcinogenicity to humans.

Line 155: Fusarium verticillioides does not produce zearalenone; it produces fumonisins. However, Fusarium asiaticum does produce zearalenone and deoxynivalenol and it could be included in the list.

Line 180: Typical producers of T-2 toxin are Fusarium sporotrichioides and F. langsethiae.

Author Response

Response to Review Comment

Dear Editors and Reviewers,

Thank you very much for giving us this opportunity to revise our manuscript, we appreciate editor and reviewers very much for your positive and constructive comments and suggestions on our manuscript entitled “Research progresses and management strategies on the common mycotoxin contamination of traditional Chinese medicines ” (Manuscript ID: jof-3598940).

We have studied reviewer’s comments carefully and have made revision which marked in blue in the revised manuscript. Attached please find the revised version, and responses to their specific comments are detailed below.

Reviewer comments:

Reviewer #1:

(1)Major comments

Overall, the authors' responses are satisfactory, and we appreciate their taking the time to review the taxonomy of mycotoxin-producing fungal species. The manuscript has been adequately revised. However, some errors should be corrected before publication (see detail comment)

Response: We sincerely thank the reviewers' valuable suggestions. We have made corresponding corrections and improvements to the errors throughout the manuscript.

(2)Detail comments

1) Line 85: I suggest the inclusion of Fusarium asiaticum as a mycotoxin-producing species.

Response: Thank you very much for pointing this out. We have classified Fusarium asiaticum as a mycotoxigenic species in line 86 of the revised manuscript.

2) Line 92: Group 3 of IARC means not classifiable as to its carcinogenicity to humans (it is not correct to say ‘due to their suspected carcinogenicity in humans’).

Response: Thank you very much for your suggestion. We have revised the manuscript content and Figure 2 accordingly in lines 92-95 of the revised manuscript.

3) Line 119: Please do not use the term Group 2B carcinogens as they are not; the correct term is Group 2B Possibly carcinogenic to humans.

Response: Thank you very much for pointing this mistake out. We have corrected "Group 2B carcinogens" to "Group 2B (Possibly carcinogenic to humans) " in line 120 of the revised manuscript.

4) Line 120: The species Penicillium viridicatum is not ochratoxin producer and must be deleted from the list. The attribution of ochratoxin A to the species P. viridicatum is based on an initial identification error that has already been clarified for some time. Misidentifying Penicillium species is common due to a lack of a complete sequence database and outdated species lists. This can lead to incorrect cultures and questionable interpretations of results.

Response: Thank you for the suggestions provided by the reviewer. We thoroughly checked the relevant literature and found that the early classification of P. viridicatum was rather chaotic. Subsequent studies revealed that some OTA-producing strains originally identified as P. viridicatum were reclassified as P. verrucosum, which is now recognized as the primary OTA-producing Penicillium species. We have deleted P. viridicatum from the manuscript in line 122 of the revised manuscript.

5) Line 153: Please do not use the term Group 3 carcinogen as they are not; Group 3 means Not classifiable as to its carcinogenicity to humans.

Response: Thank you very much for pointing this mistake out. We have corrected "Group 3 carcinogens" to "Group 3 (Not classifiable as to its carcinogenicity to humans.) " in lines 154-155 of the revised manuscript.

6) Line 155: Fusarium verticillioides does not produce zearalenone; it produces fumonisins. However, Fusarium asiaticum does produce zearalenone and deoxynivalenol and it could be included in the list.

Response: Thank you very much for pointing this mistake out. We have replaced "Fusarium verticillioides" with "Fusarium asiaticum" in line 157 of the revised manuscript.

7) Line 180: Typical producers of T-2 toxin are Fusarium sporotrichioides and F. langsethiae.

Response: Thank you for the reviewer’s suggestions. After thorough analysis, we systematically integrated the core concepts from “T-2 toxin is one type of trichothecene that is produced by some Fusarium species under specific conditions.” and “Typical producers of T-2 toxin are Fusarium sporotrichioides and F. langsethiae.” to develop the more comprehensive Sentence “T-2 toxin is a type A trichothecene mycotoxin primarily produced by certain Fusarium species, notably F. sporotrichioides and F. langsethiae, under specific environmental conditions.”.in lines 182-184 of the revised manuscript.

Reviewer #2:

(1) Major comments:

The information is presented in a very scattered and shallow manner. The arguments are not supported by data.

Response: We sincerely appreciate the reviewer’s suggestions of our manuscript. We will carefully consider your comments and further modify the manuscript.

(2) Detail comments:

  1. Detection methods of mycotoxins in traditional Chinese medicines; 4. Regulatory recommendations for mycotoxins in traditional Chinese medicines; and 5. Regulatory recommendations for mycotoxins contamination in traditional Chinese medicines

Comments:

  1. a) What is the difference between topics 4 and 5?

Response: Thanks for your comments. Due to our carelessness, Topics 4 and 5 were originally presented identically. Now we modified them, Now Topic 4 is focused on prevention and control strategies for mycotoxin contamination in traditional Chinese medicines, while Topic 5 focuses on regulatory recommendations for mycotoxins contamination in traditional Chinese medicines from a supervision perspective. Topic 4 has been revised to “Comprehensive prevention and control strategies for mycotoxin contamination in traditional Chinese medicines”. in lines 421-422 of the revised manuscript.

  1. b) Most of the data reported do not present numerical results: concentration ranges, what is the detection limit, what is the recovery of the methods?

Response: Thank you very much for your excellent comments. In the manuscript, the section on detecting mycotoxins in traditional Chinese medicines primarily focuses on identifying which mycotoxins were detected in the samples and quantifying their levels. We believe that since the relevant literature was able to determine the types and concentrations of mycotoxins, these results were inherently based on well-established methodologies. Therefore, parameters such as the linear range, detection limits, and recovery rates of the methods were not explicitly listed.

  1. c) Table 1 is incomplete: LC is not a detection method; it is a separation method. Which detector was actually used?

Response: Thank you very much for your suggestion. For liquid chromatography detection, the primary detector used is the fluorescence detector (FLD), which mainly includes two derivatization methods: post-column photochemical derivation and post-column iodine derivation. We have supplemented the relevant information in Table 1.

Thank you for your time and constructive feedback. We look forward to hearing from you soon.

With best regards.

Correspondence: Prof. Dr. Huali Xue

College of Science,

Gansu Agricultural University,

Lanzhou 730030,

China.

Tel: +86(0)931-7631212;

E-mail: xuehual@gsau.edu.cn

Reviewer 2 Report (New Reviewer)

The information is presented in a very scattered and shallow manner. The arguments are not supported by data.

The topics of the sections:

3. Detection methods of mycotoxins in traditional Chinese medicines; 4. Regulatory recommendations for mycotoxins in traditional Chinese medicines; and 5. Regulatory recommendations for mycotoxins contamination in traditional Chinese medicines 

Comments: 

a) What is the difference between topics 4 and 5?

b) Most of the data reported do not present numerical results: concentration ranges, what is the detection limit, what is the recovery of the methods?

c) Table 1 is incomplete: LC is not a detection method, it is a separation method. Which detector was actually used?

Author Response

Response to Review Comment

Dear Editors and Reviewers,

Thank you very much for giving us this opportunity to revise our manuscript, we appreciate editor and reviewers very much for your positive and constructive comments and suggestions on our manuscript entitled “Research progresses and management strategies on the common mycotoxin contamination of traditional Chinese medicines ” (Manuscript ID: jof-3598940).

We have studied reviewer’s comments carefully and have made revision which marked in blue in the revised manuscript. Attached please find the revised version, and responses to their specific comments are detailed below.

Reviewer comments:

Reviewer #1:

(1)Major comments

Overall, the authors' responses are satisfactory, and we appreciate their taking the time to review the taxonomy of mycotoxin-producing fungal species. The manuscript has been adequately revised. However, some errors should be corrected before publication (see detail comment)

Response: We sincerely thank the reviewers' valuable suggestions. We have made corresponding corrections and improvements to the errors throughout the manuscript.

(2)Detail comments

1) Line 85: I suggest the inclusion of Fusarium asiaticum as a mycotoxin-producing species.

Response: Thank you very much for pointing this out. We have classified Fusarium asiaticum as a mycotoxigenic species in line 86 of the revised manuscript.

2) Line 92: Group 3 of IARC means not classifiable as to its carcinogenicity to humans (it is not correct to say ‘due to their suspected carcinogenicity in humans’).

Response: Thank you very much for your suggestion. We have revised the manuscript content and Figure 2 accordingly in lines 92-95 of the revised manuscript.

3) Line 119: Please do not use the term Group 2B carcinogens as they are not; the correct term is Group 2B Possibly carcinogenic to humans.

Response: Thank you very much for pointing this mistake out. We have corrected "Group 2B carcinogens" to "Group 2B (Possibly carcinogenic to humans) " in line 120 of the revised manuscript.

4) Line 120: The species Penicillium viridicatum is not ochratoxin producer and must be deleted from the list. The attribution of ochratoxin A to the species P. viridicatum is based on an initial identification error that has already been clarified for some time. Misidentifying Penicillium species is common due to a lack of a complete sequence database and outdated species lists. This can lead to incorrect cultures and questionable interpretations of results.

Response: Thank you for the suggestions provided by the reviewer. We thoroughly checked the relevant literature and found that the early classification of P. viridicatum was rather chaotic. Subsequent studies revealed that some OTA-producing strains originally identified as P. viridicatum were reclassified as P. verrucosum, which is now recognized as the primary OTA-producing Penicillium species. We have deleted P. viridicatum from the manuscript in line 122 of the revised manuscript.

5) Line 153: Please do not use the term Group 3 carcinogen as they are not; Group 3 means Not classifiable as to its carcinogenicity to humans.

Response: Thank you very much for pointing this mistake out. We have corrected "Group 3 carcinogens" to "Group 3 (Not classifiable as to its carcinogenicity to humans.) " in lines 154-155 of the revised manuscript.

6) Line 155: Fusarium verticillioides does not produce zearalenone; it produces fumonisins. However, Fusarium asiaticum does produce zearalenone and deoxynivalenol and it could be included in the list.

Response: Thank you very much for pointing this mistake out. We have replaced "Fusarium verticillioides" with "Fusarium asiaticum" in line 157 of the revised manuscript.

7) Line 180: Typical producers of T-2 toxin are Fusarium sporotrichioides and F. langsethiae.

Response: Thank you for the reviewer’s suggestions. After thorough analysis, we systematically integrated the core concepts from “T-2 toxin is one type of trichothecene that is produced by some Fusarium species under specific conditions.” and “Typical producers of T-2 toxin are Fusarium sporotrichioides and F. langsethiae.” to develop the more comprehensive Sentence “T-2 toxin is a type A trichothecene mycotoxin primarily produced by certain Fusarium species, notably F. sporotrichioides and F. langsethiae, under specific environmental conditions.”.in lines 182-184 of the revised manuscript.

Reviewer #2:

(1) Major comments:

The information is presented in a very scattered and shallow manner. The arguments are not supported by data.

Response: We sincerely appreciate the reviewer’s suggestions of our manuscript. We will carefully consider your comments and further modify the manuscript.

(2) Detail comments:

  1. Detection methods of mycotoxins in traditional Chinese medicines; 4. Regulatory recommendations for mycotoxins in traditional Chinese medicines; and 5. Regulatory recommendations for mycotoxins contamination in traditional Chinese medicines

Comments:

  1. a) What is the difference between topics 4 and 5?

Response: Thanks for your comments. Due to our carelessness, Topics 4 and 5 were originally presented identically. Now we modified them, Now Topic 4 is focused on prevention and control strategies for mycotoxin contamination in traditional Chinese medicines, while Topic 5 focuses on regulatory recommendations for mycotoxins contamination in traditional Chinese medicines from a supervision perspective. Topic 4 has been revised to “Comprehensive prevention and control strategies for mycotoxin contamination in traditional Chinese medicines”. in lines 421-422 of the revised manuscript.

  1. b) Most of the data reported do not present numerical results: concentration ranges, what is the detection limit, what is the recovery of the methods?

Response: Thank you very much for your excellent comments. In the manuscript, the section on detecting mycotoxins in traditional Chinese medicines primarily focuses on identifying which mycotoxins were detected in the samples and quantifying their levels. We believe that since the relevant literature was able to determine the types and concentrations of mycotoxins, these results were inherently based on well-established methodologies. Therefore, parameters such as the linear range, detection limits, and recovery rates of the methods were not explicitly listed.

  1. c) Table 1 is incomplete: LC is not a detection method; it is a separation method. Which detector was actually used?

Response: Thank you very much for your suggestion. For liquid chromatography detection, the primary detector used is the fluorescence detector (FLD), which mainly includes two derivatization methods: post-column photochemical derivation and post-column iodine derivation. We have supplemented the relevant information in Table 1.

Thank you for your time and constructive feedback. We look forward to hearing from you soon.

With best regards.

Correspondence: Prof. Dr. Huali Xue

College of Science,

Gansu Agricultural University,

Lanzhou 730030,

China.

Tel: +86(0)931-7631212;

E-mail: xuehual@gsau.edu.cn

Round 2

Reviewer 2 Report (New Reviewer)

The authors responded to the questions raised in the previous report, but the overall text has not improved significantly. Here are some points not considered in the review.

ABSTRACT: The abstract is 11 lines long, and the expression "traditional Chinese medicine" appears 6 times. The writing is repetitive and confusing, and the abstract is superficial.

INTRODUCTION: The introduction does not contain relevant information, and, like the abstract, it is written superficially. Some examples of sentences:

 Line 30: "China is the world's largest producer and consumer of traditional Chinese medicines, as well as the largest exporter and sales market." Here, the sentence could contain additional information, such as the weight of products sold, the market value, the main purchasing countries, the most common herbs, etc. Information is non-specific.

Line 91: "Group 2B carcinogens due to their possible human carcinogen".  When the expression "Group 2B carcinogens" are mentioned, the idea is conveyed that they are a group within those classified as carcinogens. This is not the case. This expression should be reviewed throughout the text of the article. 

Line 93. "Group 3 carcinogens".  The same as mentioned above.  When the expression "Group 2B carcinogens" are mentioned, the idea is conveyed that they are a group within those classified as carcinogens. This is not the case. This expression should be reviewed throughout the text of the article. 

5.2. Supplement limit regulations

Line 476-478- " Pharmacopoeia only stipulates the limits for AFs and ZEN, and only includes 24 types of traditional Chinese medicines, which are far fewer than the actual number of medicinal materials found to be contaminated with mycotoxins. "

A table listing traditional Chinese medicines and accepted levels would add value to the text. 

CONCLUSION: The information presented in the conclusion is extremely general and could be said to apply to a wide variety of matrices subject to mycotoxin contamination. There is no relevant information that is different from what is obtained from several other publications already available. Example: "The common detection methods for mycotoxins are HPLC, HPLC-MS/MS, rapid detection technology, and field real-time monitoring technology." HPLC is a separation method. MS is an identification and quantification method. It is common knowledge that it is the most suitable method in use, and numerous review articles deal with this subject.

A review article is expected to be a compilation of clear and objective information that aggregates recent knowledge. The information contained throughout the text is confusing and truncated.

Round 3

Reviewer 2 Report (New Reviewer)

The authors made the requested corrections.

The authors made the requested corrections. 

This manuscript is a resubmission of an earlier submission. The following is a list of the peer review reports and author responses from that submission.

Round 1

Reviewer 1 Report

The whole manuscript  is written chaotically, making it difficult to understand 

• 31-32 why everything in singular? (one mycotoxin, one fungus)

• 43 would sound better “is the worlds’s largest producer and consumer”

• 58 ‘Alternatis’ or shouldn’t it be ‘Alternaria’

• 64 should add ‘,among others,’ before ‘carcinogenic’ (after all, this is not a complete list of harmful properties, e.g. OTA is nephrotoxic and ZEN has xenoestrogenic properties)

• Sentence 67-71 I can’t understand it (grammatically)

• 71 should probably be ‘on this basis’

• 90 should be ‘Fusarium toxins are’ (plural, after all, the author writes about several toxins)

• 94 should be ‘classified’

• 119 should be ‘hepatotoxic’ (the whole sentence is a double repetition of the same thing, e.g. hepatotoxic- damages liver)

• 137 should be ‘carcinogenic’

• 138 should be Fumonisins are (or Fumonisin)

• 185 should be ‘contamination’

• 192 …also known as Penicillin… ???

• 272-273 ‘All the traditional Chinese medicinal materials had different degrees of OTA pollution in natural diseases,…’ I don’t understand what the author meant

• 283-288 this whole sentence is very confusing and I don’t quite know what the author meant

• 332, 337- ‘kinds of’ is unnecessary in my opinion

• 412- ‘inadaptable’ ???

• 425- ‘very’ is unnecessary (‘critical’ already suggests it)

• In paragraph 1. The author presents the results of previous studies on mycotoxins in samples of Chinese medicinal products. Then some of these results are supplemented with a description of the method used for detection in paragraph 2. (why only some, example: publications under numbers [22] and [23])

Reviewer 2 Report

Major comments:

The review article “Research progresses and management strategies on the common mycotoxin contamination of Chinese medicinal materials ” is an interesting review about the problem of mycotoxins contamination in Chinese medicinal materials.  The manuscript reported the contamination conditions, the principal detection methods, the control measures used to contain the infection and the regulatory recommendations for mycotoxins contamination. In my opinion, the paper submitted is well written, easy to read and its structure is adequate for a review work.

To complete and improve this manuscript I have two suggestions:

-I suggest to add, at the end of introduction, the criteria chosen for references selection (keywords, years, reference types ecc)

-In my opinion, to make the review more complete, it would be interesting to add a chapter on the methods of identification of mycotoxin-producing fungal pathogens in medicinal products. For example, molecular methods of identification can also be defined as rapid indirect methods for estimating the possible presence of mycotoxins in products. The most recent work done on this topic could be added.

Detail comment:

Pag.1, line 24 and pag.2 line 58: Please, write Alternaria in the place of Alternatis

Pag.1, lines 37-38: Please, write membranaceus in italics

Pag.4, line 139: F. oxyaporum or F. oxysporum? Please, correct if it is wrong.

Pag.5, line 200: It seems to me that "Trichothecium roseum"   is not mentioned in the reference [69]. Please, if necessary, add another relevant reference.

Pag. 2, Figure 1: legend and caption are difficult to read and should be written more clearly.

Reviewer 3 Report

The manuscript contains many mistakes. Not much care has been taken with the scientific names of the producing fungi and with the names of the mycotoxins. The production of mycotoxins is attributed to fungal species that do not produce them and vice versa, and many fungal species named simply do not exist at all. The wording is confusing or ambiguous in many places, which significantly affects the scientific quality of the review.

Line 24: Typo in Alternatis (correct is Alternaria)

Lines 33-37: None of the fungal species mentioned is producer of the mycotoxins aflatoxins. In addition, if Codonopsis pilosula is a plant, why mycotoxin concentrations are expressed as ng/mL?

Lines 40-41: Please avoid using three decimal places because it creates confusion between decimal units and thousand units.

Line 58: Again, typo in Alternatis (correct is Alternaria).

Line 67-80: It is not clear what this information about the number of ‘collected AFs inspection items’ leads to. The Figure 1 is not clear enough for reading.

Line 88: What is aspertoxin? Aflatoxin producers and ochratoxin producers are mixed up.

Line 90: Fusarium moniliforme has not been called as such for more than 30 years, the correct name is Fusarium verticillioides. Fusarium pink does not exist. A review is supposed to be up to date.

Lines 92-93: Of the fungi mentioned, only P expansum is a recognized producer of patulin.

Line 123: Penicillium viridis does not exists.

Lines 138-139 and 154-155: All Fusarium species names are wrong or not fumonisin or ZEN producers.

And we could go on like this until the end of the review, but I don't think it's worth it.